# Transcriptome and Metabolome Analysis of Isoquinoline Alkaloid Biosynthesis of *Coptis chinensis* in Different Years

**DOI:** 10.3390/genes14122232

**Published:** 2023-12-18

**Authors:** Xinyi Min, Ting Zhu, Xinyi Hu, Cong Hou, Jianing He, Xia Liu

**Affiliations:** School of Chemistry, Chemical Engineering and Life Sciences, Wuhan University of Technology, Wuhan 430700, China; 17396177935@163.com (X.M.); sixtnn@163.com (T.Z.); h270718@whut.edu.cn (X.H.); hchappy1999@foxmail.com (C.H.); 294016@whut.edu.cn (J.H.)

**Keywords:** *Coptis chinensis*, RNA-Seq, isoquinoline alkaloids, biosynthesis, UPLC-MS/MS

## Abstract

*Coptis chinensis* is a perennial herb of the *Ranunculaceae* family. The isoquinoline alkaloid is the main active component of *C. chinensis*, mainly exists in its rhizomes and has high clinical application potential. The in vitro synthesis of isoquinoline alkaloids is difficult because their structures are complex; hence, plants are still the main source of them. In this study, two-year and four-year rhizomes of *C. chinensis* were selected to investigate the effect of growth years on the accumulation of isoquinoline alkaloids. Two-year and four-year *C. chinensis* were selected for metabolomics detection and transcriptomic analysis. A total of 413 alkaloids were detected by metabolomics analysis, of which 92 were isoquinoline alkaloids. *(S)*-reticuline was a significantly different accumulated metabolite of the isoquinoline alkaloids biosynthetic pathway in *C. chinensis* between the two groups. The results of transcriptome analysis showed that a total of 464 differential genes were identified, 36 of which were associated with the isoquinoline alkaloid biosynthesis pathway of *C. chinensis*. Among them, 18 genes were correlated with the content of important isoquinoline alkaloids. Overall, this study provided a comprehensive metabolomic and transcriptomic analysis of the rapid growth stage of *C. chinensis* rhizome from the perspective of growth years. It brought new insights into the biosynthetic pathway of isoquinoline alkaloids and provided information for utilizing biotechnology to improve their contents in *C. chinensis*.

## 1. Introduction

*Coptis chinensis*, belonging to the Ranunculaceae family, is one of the most widely used and valuable medicinal plants, growing in central and southern China [1]. Extracts of *C. chinensis* have been shown to have antiviral, anti-inflammatory, and antibacterial activities, as well as antihypertensive effects [2]. Modern phytochemical studies have shown that *C. chinensis* is rich in isoquinoline alkaloids, including berberine, coptisine, epiberberine, magnoflorine, columbamine, palmatine, and jatrorrhizine [3]. For artificial planting, *C. chinensis* is a slow-growing plant that generally requires a growth cycle of more than five years [4], coupled with overdevelopment in recent years, its wild resources have been endangered [5]. Because of the difference between the artificial environment and the wild environment, the growth time is bound to become a key factor in the growth and development of perennial plants. Ginseng, which also has a very long growth cycle, has a significant effect of growth years on the synthesis of ginsenosides in its cultivars [6]. In addition, alkaloids, as the most important secondary metabolites, play an enhanced role in plant stress responses to biotic and abiotic stresses [7]. Although various applications of alkaloids in *C. chinensis* have been demonstrated, the study on the regulation mechanisms of alkaloid biosynthesis in *C. chinensis* is not detailed enough.

In recent years, high-throughput sequencing technology has greatly promoted the study of plant secondary metabolite synthesis [8,9]. Based on sequencing data, key metabolites and regulatory factors are sought to elucidate their regulatory mechanisms. In *C. chinensis*, it has been reported that the P450 gene family and the OMT gene family participated in the synthesis of isoquinoline alkaloids and promoted the diversity of related BIAs. In addition, some structural genes (such as NCS, PPO, etc.) and transcription factors (TFs) (such as bHLH, WRKY, etc.) have been found to regulate the biosynthesis and accumulation of berberine [10]. Moreover, the alkaloid content of *C. chinensis* was closely related to the growth year. During the growth period, the growth and thickening of rhizomes were highly correlated with the growth years and the biomass proportion of *C. chinensis* increased year by year during the five-year growth period [11]. It has been reported that the fourth year is the fastest growth period for rhizome biomass [12]. According to the current research, the synthetic pathways of berberine, palmatine, columbamine, and magnoflorine have been very clear, while the synthetic mechanisms of coptisine, epiberberine, and jatrorrhizine are still unclear [13]. At present, because the chemical structure of alkaloids is too complex to form an economic-scale chemical synthesis, enzymes, such as biocatalysts are increasingly used as alternatives to traditional organic synthesis methods [14]. Plants are still an important source of bioactive alkaloids, therefore it is necessary to understand the regulatory mechanisms of alkaloids. In previous studies, the molecular mechanisms of the seven major isoquinoline alkaloids have been explored based on the differences in accumulation among different tissues [15], differences in plants among different regions [16], and the effects of biotic stresses on the growth and development of *C. chinensis* [17]. However, growth age is a very critical factor affecting the growth and development of cultivated *C. chinensis*.

In this study, the rhizome metabolites of two-year and four-year *C. chinensis* were qualitatively and quantitatively analyzed based on the Ultrahigh performance liquid chromatography-tandem mass spectrometry (UPLC-MS/MS) detection platform to find the key metabolites. Then, through transcriptome comparative analysis, the structural genes and transcription factors related to metabolite biosynthesis were identified. This study is helpful in elucidating the potential mechanism of the growth age on the physiological state of *C. chinensis* and prepared for further development of the value of *C. chinensis* metabolites.

## 2. Materials and Methods

### 2.1. Plant Materials

In this study, we took two-year and four-year rhizomes of *C. chinensis* for testing, which were dug from the *C. chinensis* plantation base in Yunxi City, Shiyan, Hubei Province, China (Figure 1). The surface soil of plants were washed with distilled water. The aerial parts were removed, then the rhizome was retained and quickly stored in liquid nitrogen. Two-year and four-year *C. chinensis* materials growing in the same conditions years were collected for further analysis. Samples from each year were collected with three biological replicates.

### 2.2. Widely Targeted Metabolome Analysis of C. chinensis

In order to perform a comprehensive assay of the metabolites in the rhizomes of the two groups of *C. chinensis*, the instrument used was an Ultrahigh performance liquid chromatography-tandem mass spectrometry (UPLC-MS/MS, UPLC, SHIMADZU Nexera X2, Shanghai, China; MS, Applied Biosystems 4500 Q TRAP, Thermo Fisher, Shanghai, China); the column used was the Agilent SB-18 (1.8 μm, 2.1 mm × 100 mm). Mobile phase A is ultrapure water with 0.1% formic acid, and mobile phase B is acetonitrile containing 0.1% formic acid. The following program was employed: 0–9 min, 5–95%, B; 9–10 min, 95%, B; 10–11.1 min, 95–5%, B; 11.1–14 min, 5%, B. The injection volume was 2 μL at a flow rate of 0.35 mL/min, and the column temperature was 40 °C. The triple quadrupole mass spectrometer was controlled with Analyst 1.6.3 software. The temperature of the ESI ion source was set to 500 °C and the ion spray voltage was set to 5500 V (positive mode)/−4500 V (negative ion mode). The collision gas uses a medium level of nitrogen with a multi-reaction detection mode (MRM).

Primary and secondary mass spectrometry data were subjected to qualitative analysis based on the MWDB V2.0 database, built by Metware Biotechnology Co., Ltd., Wuhan, China.

For the quantitative analysis of metabolites, after the raw data of metabolites were obtained, the mass spectrum peaks obtained were integrated and an integrated correction was performed [18]. The area of the integral was used to represent the relative content of metabolites.

### 2.3. RNA-seq Library Preparation and Sequencing

Six RNA samples (Cch2-1, Cch2-2, Cch2-3, Cch4-1, Cch4-2, and Cch4-3) were prepared using the NEBNextRUltraTMRNA Library Prep Kit for IlluminaR (NEB, Ipswich, MA, USA) following the recommendations and adding index codes to attribute sequences. Quantification and characterization of total RNA were conducted using a NanoDrop and Agilent 2100 bioanalyzer (Thermo Fisher Scientific, Waltham, MA, USA). The library preparations were sequenced on Illumina HiSeq™ 6000 sequencing platform. To obtain clean reads, Fastp was used to filter the raw data; outputs with adapters were mainly removed. All subsequent analyses are based on high-quality data.

### 2.4. Transcriptome Compared with the Reference Genome and Functional Annotation

The sequencing fragment went through a random interruption. To determine the genes from which these fragments were transcribed, the clean reads were compared to a reference genome. The reference genome is derived from NCBI (accession number PRJNA662860) [10]. Using HISAT2, the data is compared with the reference genome to obtain position information on the gene and sequence characteristic information unique to the sequencing sample. The functional annotation of the compared sequences was annotated in the Non-Redundant Protein Sequence Database (Nr), Protein family (Pfam), clusters of euKaryotic Orthologous Groups (KOG), TrEMBL, Swiss-Prot, Kyoto Encyclopedia of Genes and Genomes (KEGG), and Gene Ontology (GO).

### 2.5. Analysis of Differentially Expressed Genes (DEGs)

The expression of individual genes was calculated using features and expressed as FPKM values. Gene expression levels between different groups were analyzed using corrected *p-*values and using the Benjamini and Hochberg method. The *p-*value and |log2foldchange| were subsequently utilized to screen for significantly different genes. KEGG enrichment analysis and GO enrichment analysis were performed on the screened differential genes.

### 2.6. Quantitative Real-Time PCR (qRT-PCR) Analysis of DEGs

In order to confirm the RNA-seq results, 14 identified genes closely related to the BIAs biosynthesis pathway were selected, and 18 s (GENEBANK: DQ4068855) was used as the internal reference gene for the qRT-PCR reaction [13]. The design of primers specific to all candidate genes was conducted on Primer 5. ABScript Neo RT Master Mix (ABclonal Technology Co., Ltd., Wuhan, China) for qPCR with the gDNA Remover Kit was used to reverse-transcribe the sample RNA into cDNA. Reverse transcription was performed using 20 μL of the system mixture (containing 2 μL RNA, 5 μL 4X ABScript Neo RT Master Mix (ABclonal Technology Co., Ltd., Wuhan, China), 1 μL 20X gDNA Remover Mix (ABclonal Technology Co., Ltd., Wuhan, China), and 12 μL Nuclease-free H_2_O). The reaction procedure was 37 °C for 2 min, 55 °C for 15 min, and 85 °C for 5 min, performing only one cycle. Then, using the BrightCycle Universal SYBR Green qPCR Mix (ABclonal Technology Co., Ltd., Wuhan, China) with UDG in Applied BiosystemsTM QuantStudioTM 3 and 5 real-time quantitative PCR instruments(Thermo Fisher Scientific, Waltham, MA, USA), each sample was repeated three times. Each system contained 10 μL of BrightCycle Universal SYBR Green qPCR Mix with UDG* (ABclonal Technology Co., Ltd., Wuhan, China), 2 μL of DNA template, 0.4 μL of forward primer, 0.4 μL of reverse primer, and 7.2 μL of ddH_2_O for a total of 20 μL. The reaction program consisted of one cycle of UDG reaction at 37 °C for 2 min, followed by one cycle of pre-denaturation at 95 °C for 3 min, 95 °C for 5 s, and finally, 40 cycles at 60 °C for 30 s. The 2^−ΔΔCt^ was used to determine the relative expression of these genes.

### 2.7. Statistical Methods

All experiments were performed in three biological replicates and averages were used for representation. SPSS 26.0 statistical software was used to conduct an analysis of variance (ANOVA). A one-way ANOVA, using growth period as the factor, was employed to test the differences among gene expression levels with two-year and four-year *C. chinensis*. A Student’s *t*-test was performed to estimate *p* < 0.05.

## 3. Results

### 3.1. Widely Targeted Metabolome Analysis of C. chinensis

A total of 2074 metabolites were detected, including 556 alkaloids (19.91%), 306 phenolic acids (14.75%), and so on (Figure 2A). Based on VIP (Variable Importance in Projection) > 1 and *p* < 0.05, a total of 161 DAMs (Differential Accumulated Metabolites) were screened in two-year and four-year *C. chinensis*, of which 100 were upregulated and 61 were down-regulated. DAMs mainly included 44 flavonoids, 31 alkaloids, and 22 phenolic acid compounds. The KEGG database was used to annotate DAMs into metabolic pathways. The DAMs were annotated to the KEGG pathway, among which the pathways with significant enrichment included the metabolic pathway, ‘biosynthesis of secondary metabolites’, ‘glyoxylate and dicarboxylate metabolism’, and ’flavone and flavonol biosynthesis‘ (Appendix A). In *C. chinensis*, we focused on the “isoquinoline alkaloid pathway”. Only one DAM *(S)*-reticuline was annotated into this pathway, which is a key intermediate in the biosynthesis of different isoquinoline alkaloids. As a branch point in many isoquinoline alkaloid synthesis pathways [19], *(S)-*reticuline is transformed into different alkaloids by enzymes such as O-methyltransferase and the P450 enzyme. We used the HCA heatmap to evaluate the differences in the content of all DEMs (Appendix A) and all differential accumulation of alkaloids (Figure 2B).

### 3.2. C. chinensis Transcriptome Analysis Using RNA-seq

Based on the Illumina HiSeq™ 6000, 6 cDNA libraries of *C. chinensis* rhizomes were constructed and sequenced. A total of 292,278,660 raw reads were obtained. After using Fastap to filter the low-quality reads, a total of 280,696,630 clean reads were generated. The average Q30 was 93.40%, and the mean GC content was 42.58% (Appendix A). Clean reads were compared to the reference genome using HISAT2. A total of 199,781,459 reads and 25,910 genes were mapped on the reference genome. The comparison efficiency of transcriptome data to reference genome is 90.04–92.48%, and the comparison efficiency of each sample is higher than 90%, indicating that transcriptome data utilization is high. These data indicated that the 6 samples in this study are reliable, based on which we will conduct further research.

### 3.3. DEGs Identification and Functional Annotation

Based on the FPKM value of each gene, the DEGs were screened by setting |log2 (foldchange)| ≥ 2 and *p-*value ≤ 0.05 as the thresholds. There are 464 genes were specifically expressed in Cch-4_vs_Cch_2 (284 up-regulated, 180 down-regulated (Appendix A). Functional annotation of obtained differential genes to facilitate understanding of their biological functions, the DEGs were annotated by the GO and KEGG databases. For GO analysis, the significant DEGs were mainly enriched in the secondary metabolite biosynthetic processes (GO:004455), alkaloid metabolic process (GO:0009820), and phenylpropanoid metabolic process (GO:0009698) (Figure 3A). KEGG enrichment analysis showed that DEGs of *C. chinensis* from two different years were enriched in 113 pathways, and these pathways enriched 624 genes in total, among which the most significant 3 pathways were metabolic pathways (ko01100), biosynthesis of secondary metabolites (ko01110), and isoquinoline alkaloid biosynthesis (ko00950) (Figure 3B).

### 3.4. Identification Analysis of DEGs in Isoquinoline Alkaloids Biosynthesis

Isoquinoline alkaloid is the main bioactive component of *C. chinensis* and was mainly accumulated in the rhizome. Based on existing and reported BIAs biosynthetic pathways, we first screen structural genes using NCBI BLASTP and HMM (Hidden Markov Model). We identified a total of 768 structural genes, including 20 tyrosine aminotransferases (TAT), 8 polyphenol oxidases (PPO), 12 tyrosine decarboxylases (TYDC), 48 *(S)*-norcoclaurine synthases (NCS), 51 O-methyltransferases (OMT), 434 cytochromeP450s (P450), 24 *(S)*-coclaurine-N-methyltransferases (CNMT), 11 berberine bridge enzymes (BBE), 102 codeine-O-demethylases (CODM), and 20 tetrahydroprotoberberine oxidases (STOX). Further analysis showed that 48 of them were DEGs. Then, the method of constructing the NJ tree was used to distinguish OMT and P450 from different subfamilies: one 3′-hydroxy-N-methyl-*(S)*-coclaurine 4′-O-methyltransferase (4′OMT), three *(S)*-norcoclaurine 6-O-methyltransferases (6OMT), one *(S)*-Norcoclaurine 7-O-methyltransferase (7OMT), and four columbamine O-methyltransferases (CoOMT) were identified (Figure 4A), and one CYP719A1, one NMCH (CYP80B2), and one CTS (CYP80G2) were identified (Figure 4B). A total of 36 DEGs were identified as being involved in the isoquinoline alkaloid biosynthesis pathway (Table 1). Then, the expression patterns of these genes were evaluated using HCA heatmaps and Phylogenetic relationship analysis (Figure 5).

In order to study the correlation between the content of important alkaloids and these DEGs, a correlation analysis of these DEGs and alkaloids was conducted and the network was drawn (Figure 6). The results showed that almost all DEGs are highly correlated with *(S)*-reticuline, which also proves that *(S)*-reticuline is a key metabolite in the biosynthesis and accumulation of isoquinoline alkaloids of *C. chinensis*. In addition, magnoflorine, coptisine, jatrorrhizine, and columbamine are also correlated with almost all DEGs, berberine was correlated with *CchSTOX2*, *CchCODM2*, *CchCODM4*, *CchCoOMT2*, and *CchPPO1*, palmatine was correlated with *CchCODM3*, *CchNCS1*, and *CchNCS3,* epiberberine was correlated with *CchCODM5*.

### 3.5. Statistical Analysis of TFs

A total of 1915 TFs were identified, of which the top 10 were mTERF, NAC, C2H2, MYB, AP2/ERF-ERF, bHLH, C3H, FAR1, SNF2, SET, and bZIP (Appendix A). The families related to BIAs biosynthesis reported in *C. chinensis* were mainly bHLH families and WRKY families. In our study, 52 bHLHs and 38 WRKYs were identified. In order to study the relationship between metabolites and these TFs, a correlation network of alkaloids and TFs were performed for correlation analysis (Appendix A).

### 3.6. qRT-PCR Validation of the Isoquinoline Alkaloids Biosynthesis Related Genes

To evaluate the accuracy of transcriptome data, we randomly selected 13 genes related to isoquinoline alkaloid biosynthesis for qRT-PCR analysis. Log2 fold variation measurements were used to calculate the correlation between the RNA-Seq results (RPKM) and the qRT-PCR results (2^−ΔΔCt^) for 13 DEGs (Figure 7). The results of qRT-PCR were consistent with the transcriptome data, proving that the transcriptome data we used for subsequent analysis were reliable.

## 4. Discussion

### 4.1. Accumulation of Alkaloids in C. chinensis with Different Growth Years

Alkaloids are important secondary metabolites in plants that have been proven to have anti-inflammatory, antibacterial, and anti-tumor activities [20], and recent studies have shown that they may have the potential to treat osteoporosis [21]. The abundant pharmacological activity of alkaloids indicates that they are worthy of further study to improve yield in the future. *C. chinensis* is one of the richest natural sources of alkaloids. In previous studies on *C. chinensis* and related species, the content of several final alkaloids was detected [13,15]. In this study, we used non-targeted metabolomics to detect all the metabolites in the rhizome of *C. chinensis* in different years. It is generally believed that the growth period of *C. chinensis* is five to six years; the first and the fifth years are slow growth stages, and the second to fourth year is the rapid growth stage [11]. In the stage of rapid growth, alkaloids accumulate rapidly [11]. Our data set showed that 19.91% of the metabolites in the rhizomes of *C. chinensis* were alkaloids. During the growth period, the alkaloids in *C. chinensis* accumulate continuously with the growth years. A total of 413 alkaloids were identified; isoquinoline alkaloids are the main active ingredient of *C. chinensis*, which has been established to mainly accumulate in rhizomes [15]. In two-year and four-year of *C. chinensis* rhizomes, the contents of the isoquinoline alkaloid 8,14-dihydroflavinantine and dehassiline were the highest, indicating that they were the main alkaloids in the rhizomes of the four-year *C. chinensis*. Alkaloids are an important class of compounds in the rhizomes of *C. chinensis* and accumulate in the rhizome during the rapid growth period of 2–4 years. In previous studies, it was shown that the accumulation of alkaloids facilitates better resistance to pests and diseases in *C. chinensis* [22,23]. We hypothesize that alkaloids play a key role in rhizome growth.

### 4.2. OMT, P450 Family Contribute to Isoquinoline Alkaloid Diversity

According to existing biosynthetic pathways, many isoquinoline alkaloids are produced by the central intermediate *(S)*-reticuline [24]. *(S)*-reticuline can be used as a substrate to synthesize compounds with anti-malarial and anti-cancer activities and have been proven [25]. *(S)*-reticuline is a key compound in the isoquinoline alkaloid biosynthesis pathway in plants, including morphine and codeine biosynthesis [26], and in *C. chinensis*, including berberine, coptisine, palmatine, jatrorrhizine, columbamine, and magnoflorine are produced through the *(S)*-reticuline pathway [13]. In our research, *(S)*-reticuline was significantly up-regulated in four years of *C. chinensis*, speculated that it promotes the accumulation of isoquinoline alkaloids.

Based on the publication of the *C. chinensis* genome [10], this study used the Illumina HiSeq™ 6000 sequencing platform to screen and analyze the differentially expressed genes with different growth years, which provides the basis for mining the genes related to alkaloid biosynthesis. It has been shown that the abundance of P450 and OMT genes promotes the diversity of alkaloids of the protoberberine type [10]. Important intermediates *(S)*-reticuline for the biosynthesis of different isoquinoline alkaloids are all highly positively correlated with *Cch4′OMT*, *Cch6OMT1*, *Cch6OMT2*, *CchCoOMT1*, *CchCoOMT2*, *CchCAS1*, and *CchNMCH1*. Moreover, the expression of these genes was also positively correlated with the number of years of growth, being higher in four-year old plants, which is consistent with previous studies [11].

O-methyltransferases are a class of proteins that depend on S-adenosyl-L-methionine (SAM) for methylation reactions in various types of secondary metabolism. In combination with the isoquinoline alkaloid biosynthetic pathway, it is clear that the OMT family plays an important role in this [27]. *(S)*-norcoclaurine 6-O-methyltransferase (6OMT) is the first O-methyltransferase in isoquinoline alkaloid biosynthesis. Under its catalytic action, the hydroxyl group of norcoclaurine is attached to the S-methyl group of S-adenosyl-L-methionine (AdoMet) to form *(S)*-coclaurine [28]. The overexpression of 6OMT had a significant effect on the alkaloid content, while the overexpression of 4′OMT had a small effect on the alkaloid content. In *California poppy*, possibly due to the lack of the 6OMT-specific gene, 4OMT may catalyze the 6OMT reaction with low activity, and the production of isoquinoline alkaloids in cells with ectopic expression of 6OMT in *C. japonica* (*Cj6OMT*) transgenicity increased significantly [29]. 3′-hydroxy-N-methl-*(S)*-coclaurine 4′-O-methltransferase (4′OMT) is another methyltransferase that contributes to the synthesis of *(S)*-reticuline, and it has a different reactivity to the methylation substrate than 6OMT [30]. The 4′OMT catalyzes the conversion of 3′-hydroxy-N-methylcoclaurine to *(S)*-reticuline. Columbamine O-methyltransferase (CoOMT) was initially isolated in *C. japonica*. CoOMT catalyzes the formation of palmatine by linking the 2-hydroxyl group of columbamine to the S-methyl group of AdoMet and was demonstrated in heterologous expression experiments [31]. It has been reported that the 7OMT gene is closely related to alkaloid biosynthesis. In opium poppy stems, the silencing of the 7OMT gene resulted in a decrease in the accumulation of total alkaloids, while its overexpression significantly increased the accumulation of alkaloids in stem and leaf tissues [32]. Jatrorrhizine is a protoberberine highly related to berberine, which has detoxification, bactericidal, and hypoglycemic effects [33,34]. In previous studies, jatrorrhizine was thought to be derived from ring opening by berberine, and the enzyme that catalyzes this reaction has not been identified [34]. It is also a special proberberine with an unusual 7-O-methylation pattern, and its biosynthesis is difficult to infer from *(S)*-reticuline [35]. In *Coptis teeta*, it was verified that *Ct7OMT* mainly methylated *(S)*-norcoclaurine at the C6 site to generate *(S)*-coclaurine, accompanied by the reaction at the C7 site to generate isococlaurine [36]. It is worth noting that *Cch6OMT1*, *Cch6OMT2*, *Cch4′OMT*, *CchCoOMT1*, *CchCoOMT2*, *CchSOMT*, and *Cch7OMT1* were highly expressed in the rhizomes of four years. They may promote the biosynthesis and accumulation of alkaloids in the rhizome of *C. chinensis*.

Cytochrome P450 monooxygenases (P450s) are important proteins that are closely associated with primary and secondary metabolism [37]. Some subfamilies, such as CYP719 and CYP80, play an important role in the biosynthesis of isoquinoline alkaloids [10]. *(S)*-N-methylcoclaurine 3′-hydroxylase (NMCH) is a member of the CYP80B subfamily, and functions in the penultimate step of the pathway that generates the intermediate *(S)*-reticuline. CYP80G2 is structurally similar to CYP80A1, which is involved in intermolecular C-O phenol coupling in berberine biosynthesis, catalyzing *(S)*-reticuline to produce *(S)*-corybuberine in magnoflorine biosynthesis [38]. In the biosynthesis of coptisine, *(S)*-cheilanthifoline synthase (CFS) and *(S)*-stylopine synthase (SPS) catalyze *(S)*-scoulerine to form two methylenedioxy bridges to obtain *(S)*-stylopine, which is oxidized by STOX to obtain coptisine. CYP719A1 (CAS) contributes to the biosynthesis of berberine by catalyzing *(S)*-tetrahydrocolumbamine to form a methylenedioxy bridge to obtain *(S)*-canadine. Jatrorrhizine and epiberberine are closely related to the structure of berberine, and their biosynthetic pathways have not been fully determined. For epiberberine, there are studies showing that *(S)*-scoulerine is O-methylated at C2 [36], then forms methylated bridges and oxidizes to epiberberine by members of the OMT, CYP719, and OX families. In comparative transcription analysis, *CchNMCH1*, *CchCAS1*, and *CchCTS1* were highly expressed in the four-year-old rhizosphere, suggesting that these genes may control the biosynthesis of alkaloids during rhizome growth.

### 4.3. Candidate TFs Related to Isoquinoline Alkaloids Biosynthesis in C. chinensis

The biosynthesis of isoquinoline alkaloids is not only regulated by related structural genes but also by transcription factors such as bHLH and WRKY1 [39,40]. The bHLH family is a group of functionally diverse transcription factors widely distributed in plants and animals. In secondary metabolism, bHLH is involved in phenylpropanoid biosynthesis [36]. The first identified bHLH was derived from Catharanthus roses. Some bHLHs have been reported to be directly involved in the biosynthesis of nicotine and the biosynthesis of *C. roses* indole alkaloids in tobacco leaf plants [41]. The bHLH found in *C. japonica* was found to be involved in isoquinoline alkaloid biosynthesis, forming a distinct isoquinoline alkaloid-specific branch, unlike the MyC2-type bHLH [42].

The WRKY family is widely distributed in plants and plays an important role in physiological changes and responses to biotic and abiotic stresses [43]. *AtWRKY* has been reported to induce ectopic expression of isoquinoline alkaloid synthesis in *C. poppy* [44]. In the study of *Ophiorrhiza pumila*, *OpWRKY2* was verified as a positive regulator of camptothecin through silencing and overexpression of *OpWRKY2* [45]. In the study of *C. chinensis*, *CcbHLH001* and *CcbHLH002* can interact with promoters of genes associated with BIAs biosynthesis [46]. This evidence also shows that the alkaloid synthesis pathway in *C. chinensis* is regulated by WRKY and bHLH transcription factors. *CchWRKY2* was differentially expressed in the two comparison groups and was highly negatively correlated with jatrorrhizine and coptisine, while highly positively correlated with magnoflorine. In the future, we will carry out yeast monogamous experiments on *CchWRKY2* to verify its specific regulatory function.

## 5. Conclusions

In this study, we used the technique of combined transcriptome and metabolome analysis to screen for important metabolic pathways, genes, and metabolites. A total of 2074 metabolites and 25,910 genes were detected by extensively targeted metabolomics and transcriptomic analysis of the two-year and four-year *C. chinensis*. A key metabolite *(S)*-reticulilne and 36 differentially structured genes related to isoquinoline alkaloid biosynthesis and accumulation were identified. Among them, 18 genes were correlated with the content of important alkaloids, suggesting that these 18 genes may be the key genes in the biosynthesis and accumulation of isoquinoline alkaloids in *C. chinensis*. A WRKY transcription factor (*CchWRKY2*) that may have regulatory effects on isoquinoline alkaloid biosynthesis was identified.

These results are conducive to the further study of the biosynthesis and regulatory mechanisms of different isoquinoline alkaloids in *C. chinensis*, revealing the laws of growth and development of *C. chinensis*, and providing a certain theoretical basis for subsequent breeding and genetic engineering modification. The present study will be helpful for the improvement of the utilization value of *C. chinensis*, the effective extraction of alkaloids, and the research and development of related drugs.

## Figures and Tables

**Figure 1 genes-14-02232-f001:**
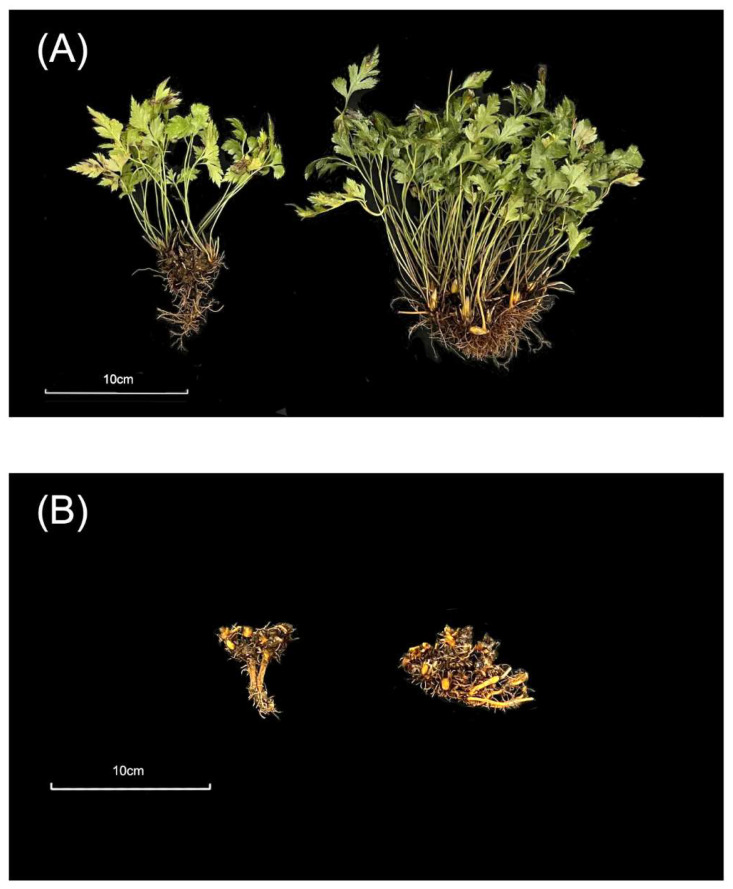
Two-year and four-year *C. chinensis* samples. (**A**) Whole *C. chinensis* plant sample. The left side was a two-year *C. chinensis*, and the right side was a four-year *C. chinensis.* (**B**) Rhizome sample of *C. chinensis*. The left sample was the two-year sample and the right sample was the four-year.

**Figure 2 genes-14-02232-f002:**
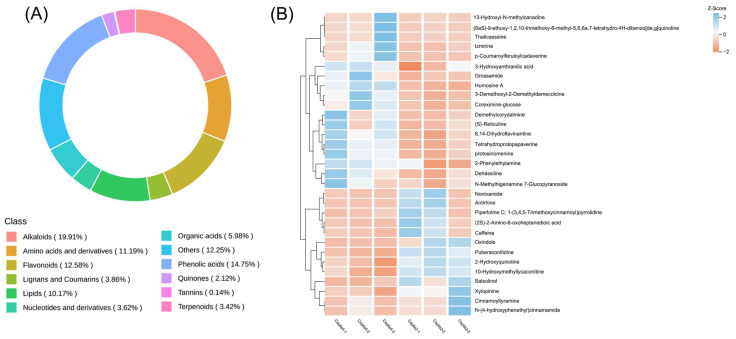
(**A**) The numbers and proportion of DAMs detected in two-year and four-year *C. chinensis.* (**B**) The content heatmap of the differential accumulation of alkaloids. Each column represents a biological repeat, and each row represents a metabolite.

**Figure 3 genes-14-02232-f003:**
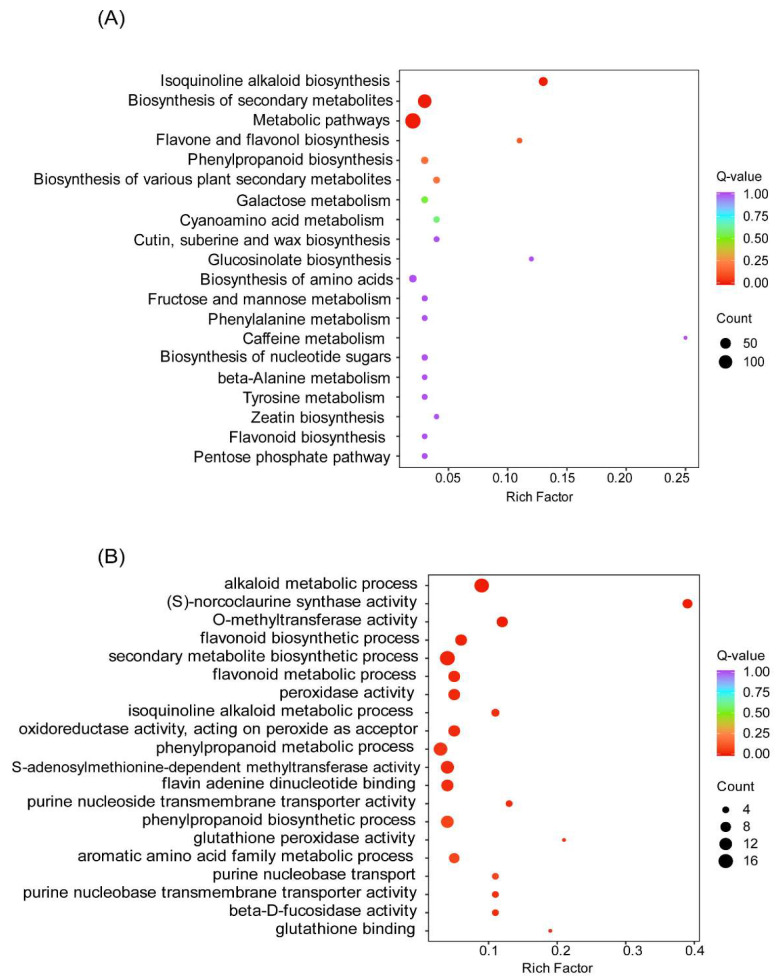
Enrichment of differential expression genes in different years of *C. chinensis.* The vertical coordinate in the graph corresponds to the biological metabolic pathway; the horizontal coordinate indicates the percentage of the number of genes enriched into each pathway; and the color of the bubbles represents the significance of the enrichment, which is indicated by the Q-value, with the more reddish color representing the smaller Q-value and the higher degree of enrichment. (**A**) KEGG Enrichment analysis of differential expression genes. (**B**) GO Enrichment analysis of differential genes.

**Figure 4 genes-14-02232-f004:**
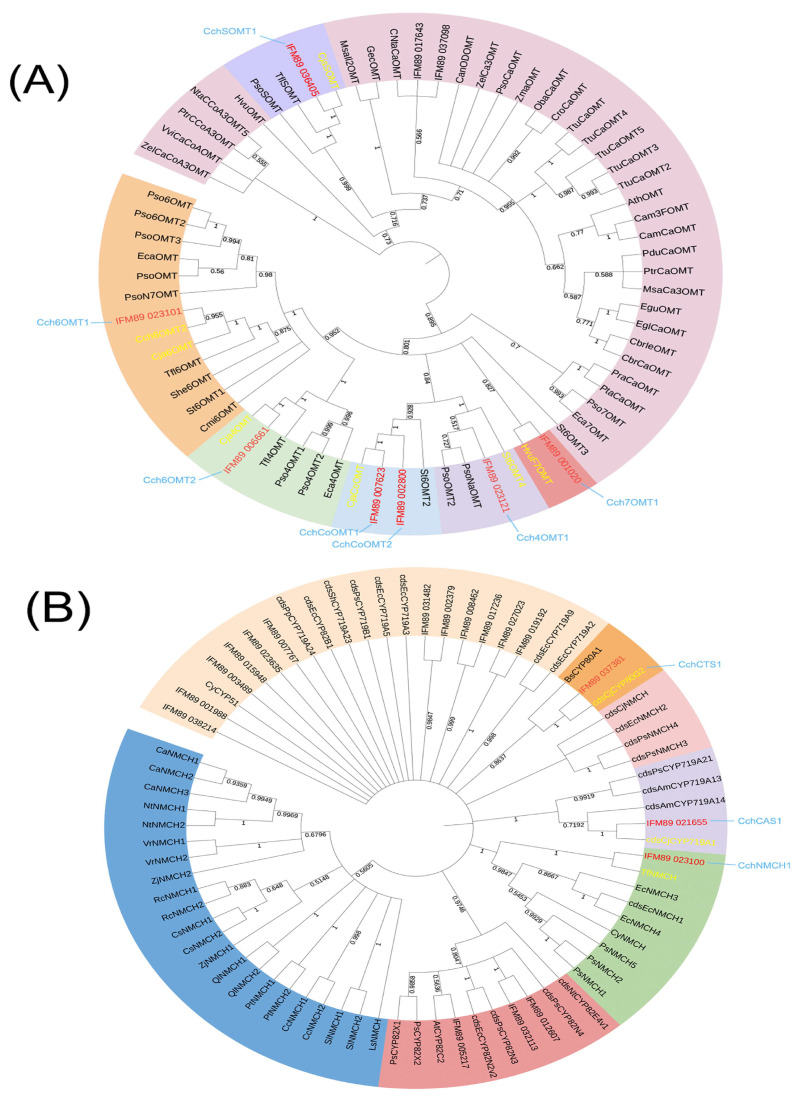
(**A**) Phylogenetic relationships of OMTs form *C. chinensis* and various other plants. (**B**) Phylogenetic relationships of P450s form *C. chinensis* and various other plants. The sequence used for the construction is shown in Appendix A.

**Figure 5 genes-14-02232-f005:**
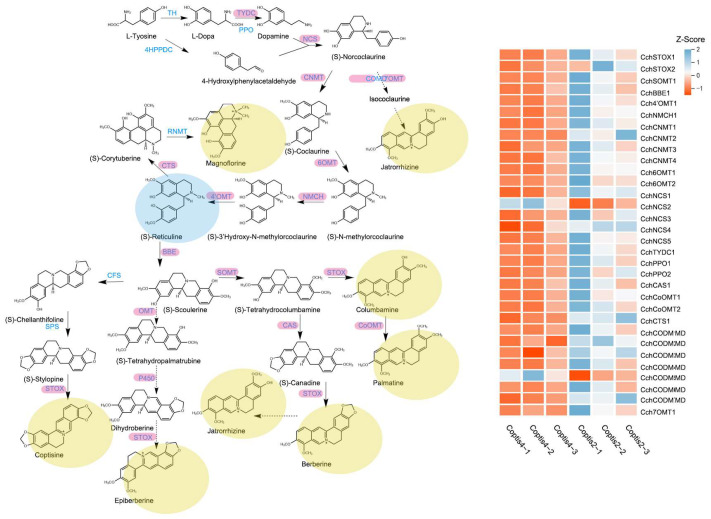
Biosynthetic pathways of isoquinoline alkaloids and expression patterns of related genes. Each column represents a group of biological repeats, and each row represents a gene.

**Figure 6 genes-14-02232-f006:**
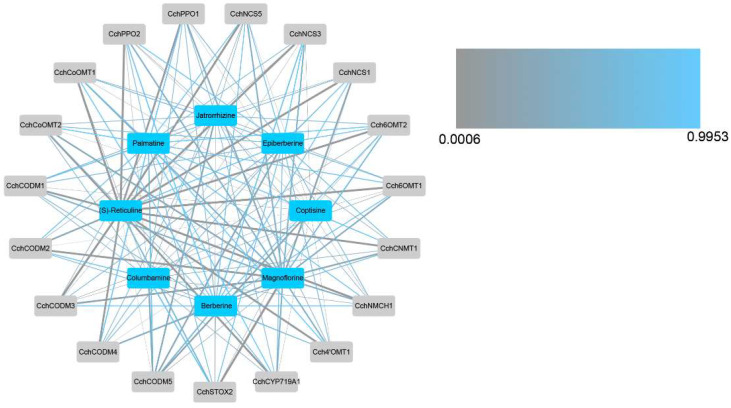
Correlation network diagram between differentially expressed structured genes and important alkaloids. The thickness of the line represents the level of correlation, and the color of the line represents the size of the *p-*value.

**Figure 7 genes-14-02232-f007:**
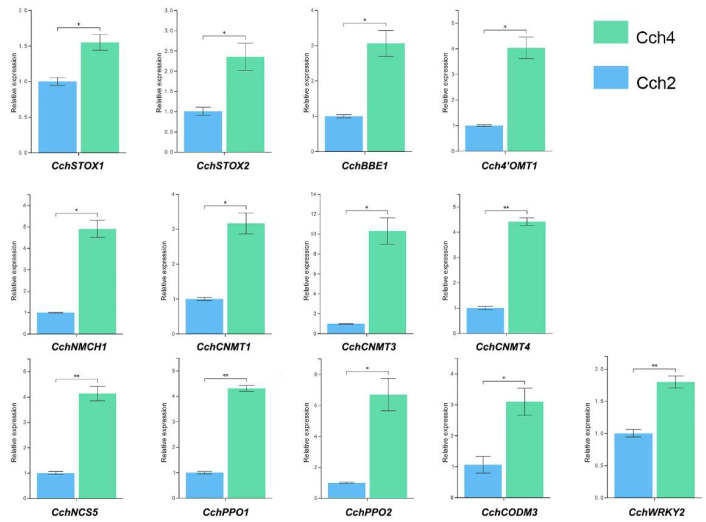
qRT-PCR results of the 13 genes related to isoquinoline alkaloid biosynthesis. The error bars represent the standard error of three biological replicates. Data were calculated using the mean of three independent biological samples, and error bars were used to indicate the standard deviation. A Student’s *t*-test was used to compare the difference between the two treatments (*n* = 3, * *p* < 0.05, ** *p* < 0.01).

**Table 1 genes-14-02232-t001:** DEGs identified related to isoquinoline alkaloid biosynthesis.

Gene Name	Gene Number	DEG Number
Tyrosine decarboxylase (TYDC)	12	1
L-Tyrosine aminotransferase (TAT)	58	0
Polyphenol oxidase (PPO)	8	2
*(S)*-Norcoclaurine synthase (NCS)	48	5
*(S)*-Norcoclaurine 6-O-methyltransferase (6OMT)	51	3
*(S)*-Norcoclaurine 7-O-methyltransferase (7OMT)		1
3′-Hydroxy-N-methyl-*(S)*-coclaurine 4′-O-methyltransferase (4′OMT)		1
Columbamine O-methyltransferase (CoOMT)		4
*(S)*-Scoulerine 9-O-methyltransferase (SOMT)		1
*(S)*-Coclaurine N-methyltransferase (CNMT)	24	4
Berberine bridge enzyme (BBE)	11	3
*(S)*-Tetrahydroprotoberberine oxidase (STOX)	20	3
Codeine-O-demethylase (CODM)	102	7
*(S)*-Cheilanthifoline synthase (CFS)	434	0
*(S)*-Stylopine synthase (SPS)		0
*(S)*-N-Methylcoclaurine-3′hydroxylase (NMCH)		1
*(S)*-Canadine synthase (CAS)		1
*(S)*-Corytuberine synthase (CTS)		1
Total	768	36

## Data Availability

The original contributions presented in this study are included in this article/in the Appendix A, and further inquiries can be directed to the corresponding author.

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
