# Peer review of "Transcriptome and Metabolome Analysis of Isoquinoline Alkaloid Biosynthesis of Coptis chinensis in Different Years"

_genes, 2023, doi:10.3390/genes14122232_

Round 1

Reviewer 1 Report

Comments and Suggestions for Authors

This article investigated the metabolome and transcriptome data of isoquinoline-related biosynthesis pathways based on MS, RNA-seq, and qRT-PCR technology. Some minor issues should be fixed/answered.       

1.       Line 40, what is the ‘abiotic organisms’?

2.       Please correct the language issues in lines 58, 79, 81, other similar issues should be avoided in the rest text.

3.       In figure 2B, how did you define the heatmap’s color? If the color represents the abundance of the metabolites: the levels of six metabolites in the triplicates are so different from each other, how could persuade the readers that these data are representative and reproducible?    

4.       Line 285-287, this statement is not clear, can you give a further demonstration on its causality?

5.       Line 293, “morphine” and “codeine” are compounds or pathways here? This sentence is confusing.

6.       What is Q-value mean in figure 3?

7.       Statistical method should be given in the method section.

Comments on the Quality of English Language

Some atttention should be paid to the corner of lanuguage in this manuscript.

Reviewer 2 Report

Comments and Suggestions for Authors

The Manuscript entitled Transcriptome and Metabolome Analysis of Isoquinoline Alkaloid Biosynthesis of Coptis Chinensis in Different Years deals about transcriptomic and metabolomic analysis about isoquinoline alkaloids.

The manuscript need some revisions:

I didn’t find Figure S1, S2 and S3.

Many abbreviations (BIAs, etc) lack.

You should better explain how you carried out the qPCR, reference genes used, relative expression method, etc. in material and methods. The results obtained about relative expression have not been well described in the Results section. In the discussion you discuss the result only for CchWRKY2 gene, although results obtained are important.

I don’t understand reference genes utilized, only 18s?

In the discussion section (4.2 OMT, P450 Family….) you widely discussed about the data of previous studies, you should focus about the similarities or differences with previous literature.

In the conclusion you should add a sentence about the biotechnological applications.

Comments on the Quality of English Language

The English must be moderately revised

Reviewer 3 Report

Comments and Suggestions for Authors

This study explored how the growth years (two-year and four-year) of Coptis chinensis rhizomes affect the accumulation of isoquinoline alkaloids. Metabolomics detected 413 alkaloids, including 92 isoquinoline alkaloids, highlighting (S)-reticuline as a significantly different metabolite in the biosynthetic pathway between the two age groups. Transcriptome analysis identified 464 differential genes, 36 of which related to isoquinoline alkaloid biosynthesis, with 18 genes linked to important alkaloid content. This comprehensive analysis sheds light on isoquinoline alkaloid biosynthesis, offering insights for leveraging biotechnology to enhance alkaloid levels in C. chinensis.

However, here are some comments below :

The introduction is well written and gives the reader an overview of the topic.

lines37-38 Comparison with other plants should be moved to the Discussion section

54-61: The objectives are not well highlighted.

62-72: This paragraph belongs to the next section, Material and Methods.

The Materials and Methods section provides a clear and detailed description of the experimental design, procedures, materials, and analytical methods used in the study: Plant Materials, Widely Targeted Metabolome Analysis of Coptis chinensis, RNA-seq Library Preparation and Sequencing, Transcriptome Compared with the Reference Genome and Functional Annotation, Analysis of Differentially Expressed Genes (DEGs) and  Quantitative Real-time PCR (qRT-PCR) Analysis of DEGs.

The Results section presents the findings of the study based on the analyses performed.

154-155 lines belong to Material and Methods 

168- revise the bibliographic reference as required 

Table and figures are clearly labeled, with detailed captions explaining their significance.

Discussion

273-274 I agree with this statement but it does not belong here. Future studies should be highlighted at the end of this section or, better, in the next section, Conclusions.

272-273 what are these previous studies? add bibliographical references

273-274 'In this study, we used untargeted metabolomics to detect all metabolites in the rhizome of C. chinensis in different years': I do not understand the role of this statement in this section.

275-276 add bibliographical references

298 add bibliographical references

Conclusions are clear and concise.

However, more bibliographical references are needed.
